# Social Innovation to Sustain Rural Communities: Overcoming Institutional Challenges in Serbia

**Ivana Živojinović [1],\*** **, Alice Ludvig [1] and Karl Hogl [2]**

1    Institute of Forest, Environment and Natural Resource Policy, University of Natural Resources and Life Sciences, Vienna (BOKU) and European Forest Institute, Forest Policy Research Network, Feistmantelstrasse 4, 1180 Vienna, Austria; alice.ludvig@boku.ac.at
2    Institute of Forest, Environment and Natural Resource Policy, University of Natural Resources and Life Sciences, Vienna (BOKU), Feistmantelstrasse 4, 1180 Vienna, Austria; karl.hogl@boku.ac.at
\*    Correspondence: ivana.zivojinovic@boku.ac.at

**Abstract:** Responding to a number of longstanding challenges such as poverty, wide-ranging inequalities, environmental problems, and migration, requires new and creative responses that are often not provided by traditional governments. Social innovations can offer socio-ecological and economic solutions by introducing new practices that reduce social inequalities, disproportionate resource use and foster sustainable development. Understanding the role of social innovations is especially complicated in unstable institutional environments, e.g. in developing countries and countries in transition. This paper analyses nine social innovations in rural areas in Serbia, based on in-depth interviews and document analysis. This analysis reveals factors that facilitate or constrain social innovations whilst simultaneously identifying related formal and informal institutional voids, for example, poor law enforcement, a lack of adequate infrastructure, lack of trust, as well as norms and values that bolster patriarchal systems. The results that emerged from this research show that social innovations are operating in spite of these challenges and are facilitating improvements in a number of the aforementioned challenging areas. Some innovators engage in social entrepreneurship activities because of subsistence-oriented goals, while others follow idealistic or life-style oriented goals, thus creating new social values. Moving beyond these observations, this paper also identifies means to overcome institutional voids, such as creation of context-specific organisational structures, improved legal frameworks, and innovative financial mechanisms.

**Keywords:** institutions; policy support; institutional void; transition countries; forestry; rural development

---

## 1. Introduction

Societies around the world are facing a great number of complex and longstanding challenges such as poverty, hunger, increasing inequalities in different spheres of life, environmental challenges, and unprecedented levels of migration [1]. It is becoming increasingly apparent that solutions for such pressing problems cannot be addressed solely by traditional governmental approaches as they are not delivering the required policy results [2]. This weakening of state capacity to deal with these issues has been accompanied by evolution within civil society that has seen the emergence of new citizen-actors and new forms of mobilisations [2] which find innovative ways to fill these capacity shortfalls. In such situations, where there are no generally accepted rules and norms about how policymaking and politics are to be conducted, we can talk of existing institutional voids.

The existence of institutional voids means that an institutional framework necessary to guide and support the proper functioning of activities within a given context is absent, weak or deficient [3,4]. Such voids stem from information problems as well as misguided and inefficient regulatory implementation

mechanisms [3], but may also include the lack or failure of informal institutions [5]. Such conditions constrain and impede solving specific policy issues [6] and lead to severe social inequalities [5]. In these conditions, multiple actors (companies and other types of organisations) seek innovative solutions to mitigate social problems [7,8].

In such contexts, social innovation has gained interest among policymakers, foundations, and researchers largely because they are assumed to offer solutions to not just localised problems but also to systemic and structural ones [9]. In policy discourse, social innovations have been presented as a solution to many kinds of old and new social challenges at a time when there is growing economic pressure on public administrations [10,11]. Social innovations are seen as an opportunity to support social wellbeing [12,13], tackle marginalisation [14], and trigger transformative changes through collective action [15].

Numerous definitions of social innovation have been proposed over time [11,16–21]. Some of them focus on new actor-relationships, interactions and new decision-making processes, whilst other definitions arise from having their focus elsewhere [22]. What is common to most of them is that they focus on various "new arrangements" to address societal needs and problems [23]. For the purpose of this research, we apply the following definition developed by the SIMRA consortium which states that social innovation is "the reconfiguring of social practices, in response to societal challenges, which seeks to enhance outcomes on societal well-being and necessarily includes the engagement of civil society actors" [13] (p. 1). We understand social innovation as a broad process, encompassing the concepts of social entrepreneurship and social enterprise, but also a range of other social initiatives and activities, which can be identified as socially innovative in certain settings. We stress this issue to clarify that we do not restrict the term "social innovation" to the establishment and activities of social enterprises, not all of which are innovative [7]. As such, this study focuses on institutional settings that are of relevance for various types of social innovations and enterprises within the Serbian context.

*The Role of Social Innovations in Rural Areas*

There is agreement that the contribution of social innovations is bringing positive change that can influence the overall development of urban communities [24]. However, the potential of social innovations has also been acknowledged and studied in the context of rural areas [8,23,25–30]. Developing new arrangements and cooperation modes for specific regions and local problems can support rural communities in their efforts to address their current challenges as well as contribute to reducing social inequalities and disproportionate resource use. Many studies have shown that the common aim for engaging with social innovation is an increased sense of belonging to a local area and community and the desire to prevent excessive emigration [28] by creating innovative and vibrant rural societies [16].

As noted by Copus et al. [8] social innovation is simultaneously dependent on local resources and participation on the one hand, and interconnections across geographical and organisational boundaries on the other. Such a multi-stakeholder perspective emphasises the importance of links between civil, public and private sector actors and the reinvention of the traditional roles of actors, which is seen to be crucial for rural social innovation processes [8,31]. Studies that explore how social capital stimulates social innovations and entrepreneurship in rural areas show that the complex interplay of different forms of social capital is important for developing socially innovative initiatives [32–35]. Lang and Fink [32] emphasise the complexity of the intermediary role of social entrepreneurs, linking local communities to powerful regime actors in multilevel network arenas within rural contexts. This also shows that social innovations are related to the much broader and dynamic political-economic context. Understanding how social innovators are impacted by the institutional framework in the process of generating ideas and solutions is important for effectively improving any institutional environment. This is especially important in unstable and deficient institutional environments, e.g. in developing, war-ravaged, or transition countries—where policy problems need to be solved despite significant institutional voids [3]. One strand of literature assumes that new socially responsible initiatives need

strong and functioning institutional arrangements, where the government, markets and civil society create and maintain an enabling environment for innovations [36–38]. However, another strand of literature empirically shows that social innovation initiatives usually emerge in environments that are institutionally deficient [7,39–42].

This paper aims at examining the assumption that institutional voids impede the contributions of social innovations to sustain and develop rural communities in Serbia. For this purpose, we focused our research on both supporting and hindering factors which influenced the selected case studies of social innovations. We assumed that by elaborating a deeper understanding of the respective innovation processes we will also identify institutional voids. For this, we aimed to provide answers to the following two research questions:

- What are the particular institutional voids that hindered the emergence and development of social innovations?
- Which supportive factors are helping to overcome identified institutional voids?

This paper contributes to the research field of social innovation in rural areas, more specifically, in rural areas of countries in transition. In doing so, it explores empirical evidence in case studies and draws on experiences of social innovators from Serbia, a country with an economy in transition. The research field of this contribution is still in an embryonic phase. Only a few articles have currently been published that investigate how institutional factors affect and shape social innovation processes in the fragile contexts of emerging market economies [7,40–42]. In fact, there are no empirical studies involving social innovation in rural sectors in Serbia. This article serves to fill this gap.

Following the introduction, we first outline the theoretical background by introducing an institutional void perspective. Section 3 begins by describing the research design as well as the methods of data collection and analysis as applied to our case studies before introducing the institutional context of social innovations in Serbia. The following results section, Section 4, is organised into three parts: part one provides descriptions of our nine case studies, based on the data collected; part two identifies factors that support social innovations, and part three describes those factors that were found to hinder the social innovations researched in our collection of case studies. The Section 5 discusses our findings from an institutional void perspective. Finally, the concluding section demonstrates that there is a pressing need to improve the institutional context in Serbia to better support and sustain social innovation initiatives in the long run.

## 2. Theoretical Background

We built our research on institutional theory, recognising that human behaviour is shaped jointly by the constraints, incentives and resources provided by formal and informal rules (institutions), which can be more or less compatible with each other [43]. Institutional theory defines institutions as humanly devised rules that structure political, economic and social interactions [44]. Institutions can be formal and informal [44,45]. In the category of formal institutions, we understand these to be the set of regulatory institutions, such as laws, regulations, strategies, as well as the constraints and incentives arising from government regulations. Informal institutions refer to more implicit, slowly changing, culturally transmitted and socially constructed rules of behaviour. These can be further divided into cognitive and normative institutions [45] and represent more tacit constraints on societies which guide expectations and ensure greater predictability in social exchanges thereby shaping individuals' and organisations' choices and actions [44]. Guided and facilitated by institutions, profit and non-profit entities provide numerous products or services designed to respond to social needs which are not always addressed by institutions [46]. Similarly, innovations occur under the influence of existing institutional environments and their success or failure is largely determined by this influence [46].

Examining social innovations, most of the literature focuses on social innovations in developed countries in Western contexts [47]. Such cases represent situations markedly different from those in developing or transition economies, where poverty, unemployment and diverse social problems are much more pronounced [7,40–42] and are characterised by institutional voids [3,4].

As is the case with the division of institutions into formal and informal, there is also a division of institutional voids into formal and informal. Formal institutional voids assume a lack of or failure of formal institutions (i.e., laws, regulations, infrastructures, and supporting apparatuses) to facilitate efficient and effective market transactions and operations [3]. Greater stability in and efficacy of formal institutions in principle can better support entrepreneurial activities [46,48]. Formal institutional voids can take different forms, manifesting themselves as ill-defined regulations, a lack of well-defined property rights and minimal investment sources provided by the state or the absence of or poorly developed infrastructure. Other voids relate to the lack of formal educational organisations which lead to a pool of unskilled potential employees. Furthermore, inadequate provision of specialised information, non-participative procedures by governmental bodies, coupled with the absence or non-functional institutionalised intermediaries are also types of such institutional gaps [46,49,50]. Such voids can hamper development by prohibitively increasing operational costs or favouring one segment of the population over another [5,46]. These voids can differ even within a country because the implementation of formal rules can vary across regions [51]. This especially becomes relevant when we focus on rural areas, which do not have the same support in terms of infrastructure and other resources when compared to urban areas.

It is assumed that informal institutions can compensate for the deficiencies resulting from a formal institutional void [5,52]. However, there may also be informal institutional voids, i.e. a lack of or a failure of informal institutions to support efficient and effective market transactions. This conceptualisation does not refer to missing norms, values and beliefs, but to settings in which there is a lack, suppression, or limited manifestation of very specific informal institutions that could support efficient and effective market transactions [5]. Informal voids exist for a number of reasons, can differ significantly between localities [5] and can be dependent on the durability of informal institutions [53]. For example, in a number of countries' patriarchal-based systems women have been excluded from participation in economic activities and lack access to property rights. Such voids create barriers for women to build personal financial security and negatively influence their ability and/or willingness to invest in a business [46]. This is social exclusion or marginalisation, which stems from norms and beliefs in a society that certain individuals, based on their gender, ethnicity, age, or other demographic attributes, lack the status to take part in certain market activities, to own property, and/or to participate in certain activities [54]. Other informal institutional voids may also exist when dominant societal beliefs allow elites to leverage their own power and misallocate public resources that satisfy their own personal interests rather than supporting efforts that further broad local development. Because of their socially accepted status, elites may also be allowed to ignore other norms to the detriment of other groups [46]. Additionally, various technological advancements impacting a society (i.e. climate science, transportation, and medicine) can be met with varying levels of institutionally-derived scepticism. This often becomes manifested with the favouring of ineffective practices or applying traditional management that can hinder the developmental processes of enterprises or result in ineffective use of resources [46]. Other observable informal institutional voids include relationship barriers arising from a lack of trust in society for various reasons [46], e.g. because of corruption [46,55].

## 3. Methodology and Methods

### 3.1. Research Design, Data Collection and Analysis

Since the aim of this study is to provide in-depth analyses of social innovations and related institutional voids, we applied a qualitative research design to yield thick descriptions of processes of social innovation in a multiple case study research design. The first part presents factors that enabled

or hindered the development of social innovations in Serbia and summarised characteristics of the selected case studies. The second exploratory stage [56] was designed to provide a better understanding of the institutional void concept in this particular setting and related supporting institutional factors that to a greater or lesser extent succeed in filling those gaps. In this study the term "hindering factors" encompasses a broad scope of factors that are or create barriers for social innovations. However, it must be noted that such hindering factors are not necessarily manifested as or even result from an institutional void.

For data collection purposes several qualitative techniques were applied, such as in-depth interviews, content analysis of organisations' websites and other materials and a literature review. Triangulating data collection by applying these various techniques allowed us to cross-check data validity and reliability. Primary data was collected by conducting nine qualitative in-depth interviews, with the key representative of each social innovation case study, i.e. with those people who initiated and further developed the respective innovation. They provided extensive knowledge of the cases, the respective context and individual perspectives. In order to understand the meanings that the respondents assign(ed) to institutional factors, an interview guide with nine open questions was developed. Interviews were conducted from September 2018 to February 2019 and lasted from 45 to 90 minutes, with an average length of 60 minutes. They were conducted face to face, in the Serbian language and recorded. Secondary data was collected by content analysis of various organisations' websites and materials as well as a literature review of publications on social innovations and entrepreneurship in Serbia, inter alia for gaining a richer understanding of the contextual factors when undertaking the case study analyses.

Data analysis started with a transcription of interviews in the NCH software. Subsequently, we applied an analysis of the transcripts by inductive coding, assisted by Atlas.ti software. Through an iterative process, initial codes were grouped into more focused and substantive categories of supportive and hindering factors, which are of relevance to the particular case studies (see Appendix A) and these codes were then related to concepts of formal and informal institutional voids.

### 3.2. Case Study Sampling

All case studies selected for this research are located in Serbia, which was chosen for this research as it is a rich empirical site where social innovation discourse has emerged in the recent years with an increasing number of social innovation initiatives. Due to its historical background and current transition phase, Serbia was viewed as highly relevant for this research not only because of its overall socio-economic system but also because of its accession process for EU membership and its resultant processes of harmonisation, legislation, and regulation.

The selection of social innovation cases for analysis was done by purposive sampling. Cases were identified through various methods—initial desktop research and screening, and expert's interviews with actors active at the national level [57]. Some cases were identified by the snowball technique (innovators suggesting other cases) [58].

The selection of the nine case studies was based on the following criteria: (1) "innovation" in terms of our definition of social innovation [13] (p. 1), (2) being active in rural areas, (3) representing different types in terms of organisation and size. We also desired to have cases with a broad geographical distribution across Serbia in order to identify specific challenges occurring in different local contexts.

### 3.3. Background Descriptions

After Serbia's democratic change in 2000, the economy and provision of social services virtually collapsed after decades of state control [59]. Serbia entered a transition process requiring profound economic and political reforms that took important steps toward firmly establishing democracy and a functional market economy. The initial steady growth rate was stopped with the financial crisis in 2008, from which the country is still struggling to recover. Currently, the most pressing social problems are widespread poverty, rising unemployment, regional disparities combined with corruption and

inefficient public administration [59,60]. The high unemployment rate is one of the largest problems facing the Serbian economy with some 14.8% of the workforce being unemployed in the first quarter of 2018 [61]. The widespread poverty and social problems, together with limited support from both the private and the public sector, has created a need for innovative models that could support recovery and growth, bringing further economic reform and positive social change [59].

The private sector started to develop significantly after the 1990s with the process of economic transition and after the cessation of state-owned enterprise activities in many sectors [62]. However, socialist and post-socialist governments have not encouraged and supported an entrepreneurial culture [59,62], meaning there is still genuine reluctance on the part of the population when it comes to starting private businesses. As such, the level of development of the private sector in the Serbian economy is still low, even in regional comparison [63].

In the past decade, social economy models were introduced [59] which stimulated the emergence of social enterprises and recent social innovation concepts, often mixed with social entrepreneurship concepts, have become prominent in debates in the last few years. Despite limited institutional support and recognition of such concepts by policymakers, there is a growing interest in social innovation and social entrepreneurship in Serbia, mainly by civil society organisations. Social enterprises and innovations are emerging in an evolving institutional framework without targeted support or specialised public sector partners [64]. A major factor driving interest in this social innovation and entrepreneurship is the accession process to the European Union and the large inflow of European and international funds [63].

Some researchers argue that the social entrepreneurship sector in Serbia has entered its institutionalisation stage with the introduction of social cooperatives as a category with the law on cooperatives and the recognition of social enterprises as service providers in social care [64–67]. Since 2012, there have been several attempts to also pass the Law on Social Entrepreneurship by the Ministry of Labor, Employment, Veterans' and Social Affairs, but the various drafts have thus far not met the expectations of stakeholders. A major point of disagreement has been that the law restricts the social entrepreneurship concept to the employment of vulnerable groups, puts an obligation to the organisations to transfer 50% of their profits into a state fund and limits the level of profit that social enterprises can make. When considering the broader perspective, social innovation in Serbia is regulated primarily by a number of policy documents stemming from different sectors and there is clearly a lack of strategically oriented and legally-binding legislation supporting social innovations [57].

## 4. Results

The analysis of nine cases of social innovations in rural areas in Serbia provides a very rich and manifold picture of how various factors can support or hinder the development and existence of these initiatives. This paper focuses on the identification of institutional voids, which are revealed by looking at hindering factors and identified needs. This, together with the overview of supporting factors, serves as a solid foundation for a better understanding of how social innovation cases are coping with and addressing institutional barriers or voids, and best facilitate drawing conclusions about potential solutions.

### 4.1. Overview of Cases

The nine cases represent examples of social innovations initiated by civil society organisations. They started operations after the year 2000, i.e. following the start of the transition process in Serbia.

> " . . . after the 2000s when we chose our European perspective, which I cannot grasp for life of me, there was a period of 10 years of complete idiocy and a rural policy without ideology and clear aim . . . that only served to reduce the attractiveness of the agri-business sector in order to privatise it to big companies. And when this was done, sometime in 2011, the state realised that our young people were in exodus, especially from the village, it was total devastation of rural areas . . . " (CS7)

It was against this backdrop and the environment it had created that the social innovations we used as our case studies came into being. Our nine case studies represent examples of social entrepreneurship and innovation development in various rural areas of Serbia. The geographical distribution provides different contextual and institutional conditions for developing social innovations [64] given the variations in infrastructure availability and development between the northern (autonomous province of Vojvodina) and the southern part of Serbia, which is more under-developed and is rather isolated and marginalised. This is connected to the socio-economic power of the regions, where the north, west and central parts of Serbia feature more stable economic conditions and higher involvement of various actors. For example, in the province of Vojvodina, the presence of the regional government, as well as the number of organisations who work and support minority groups, provides a much stronger portfolio of economic and other types of support for development. This, at the same time, is challenging as it leads to weaker support from the central government (CS1, 8 and 9). A further difference between the regions is found in terms of natural resource potential. Figure 1 presents the geographical distribution of the cases selected for this study.

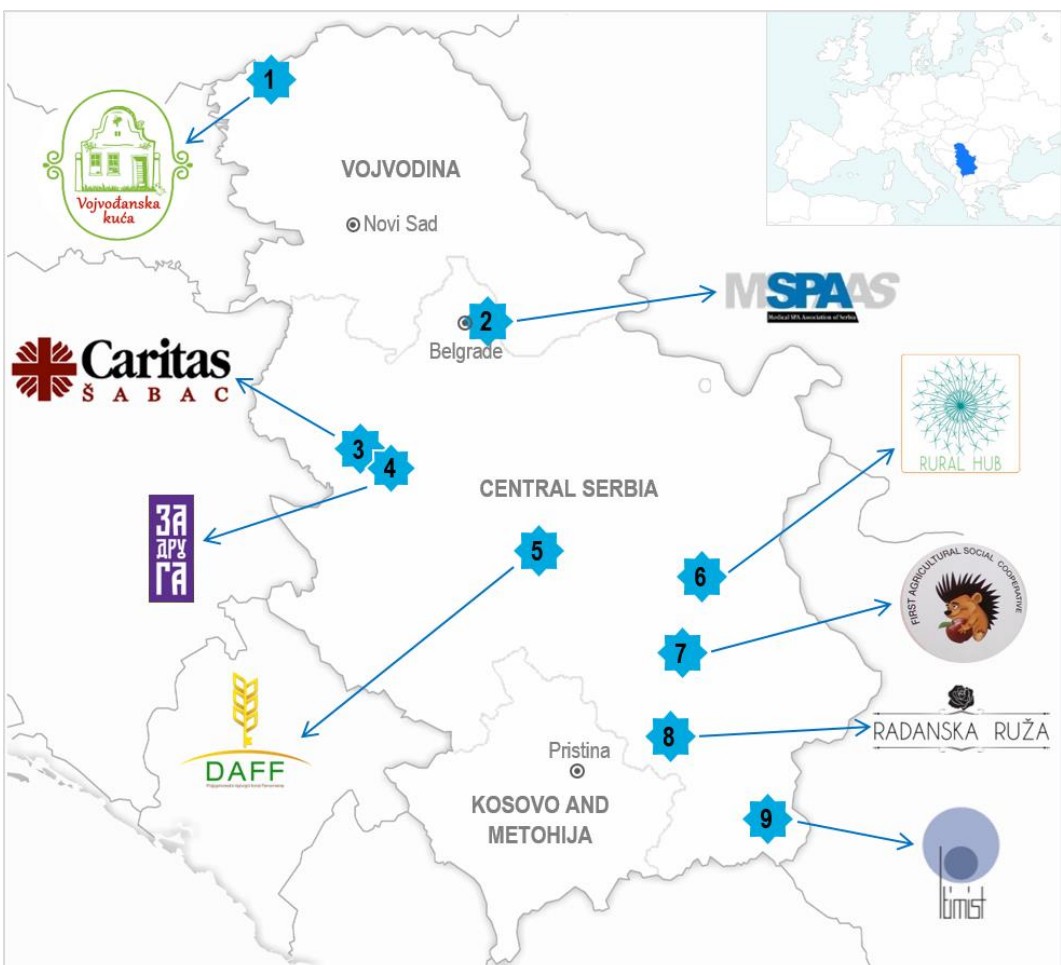

**Figure 1.** Geographical distribution of case studies (created by the authors).

These social innovations are tightly linked to the initial work of initiating associations and their work with various vulnerable groups such as people with disabilities (CS 3 and 8), unemployed youth (CS 7), women victims of violence or human trafficking (CS 1, 5, and 9), the Roma population (CS 9), efforts to address social injustice (CS 4), gender equality (CS 1, 5, 8, and 9), rural development (CS 6) and nature-health issues (CS 2). Through their previous work, these associations identified many problem areas which their local communities face and at the same time they identified opportunities to

improve local living conditions, often by providing work opportunities, or by creating conditions in the villages that will, in the long run, prevent emigration and contribute to the sustainable development of their initiatives.

> " ... we plan that what we do has multiple effects on the community, to also have an environmental impact, a social impact, to look at society from a broader perspective ... " (CS5)

In terms of formal organisational structure, some of these are registered as associations, under the Law on Associations (2009) (CS1, 2, 3, 4, 5, 6, 9). Some of them are undertaking economic activities according to the same law, which allows the formal registration of the economic activities of associations in order to provide additional resources necessary for carrying out their basic (non-profit) activities (CS1, 3, 4, 9). The "Radanska Ruza" initiative (CS8) is another example, of a non-profit limited liability company that is a public-private partnership between "Women's Association Ruza" and the municipality Lebane. The "First Social Agricultural Cooperation" (CS7) is operating under the Law on cooperatives (2015), which for the first time recognises "social cooperative" as a formal type of organisation.

A majority of our cases started from the formation of social enterprises with production-oriented activities primarily in the field of organic agriculture (CS1, 3, 4, 7, 8) or non-wood forest products (CS9). Pursuing such economic activities was designed to secure the financial and long-term sustainability of the work of the association and reduce their dependence on donors' funds (chiefly international funds) whose initial support enabled all the organisations to be established in the first place. The profit generated from these activities is reinvested by the associations in social projects of their own or other similar organisations aimed at enhancing the wellbeing of their local communities. Apart of the economic activities, each of these organisations are creating different partnership models with the users and producers, thus giving these latter two groups equally important roles in the various social innovation aspects. Additionally, the organisations are actively working on educating and improving the skills of their producers/partners, frequently in partnerships with other civil society organisations (e.g. business plan development, branding, marketing, etc.).

In other case studies, social innovation is provided through the active involvement in the improvement of living conditions in local communities, by taking a participative approach for involving the communities in the work of the organisations. The case of "RuralHub" (CS6) works on empowering a village population by connecting rural and urban values and producing a portfolio of activities which succeed in activating and promoting the village, e.g. creating co-working space, the production of traditional products, conducting educational programmes, and improving tourism opportunities. The central role of the community is illustrated by the following quote from a "RuralHub" founder:

> "What is innovative about RuralHub is the community, where the Hub is the whole village" (CS6)

The "Fenomena" association (CS5) with its Development Agriculture Fund Fenomena (DAFF) works on raising funds for individual agricultural producers in the region who are selected on very strict socially-oriented criteria. Thus, it integrates marginalised groups within local society into the market and seeks to prevent their exodus from the area. The "Medical Spa Association" (CS2) is socially innovative because it works to connect various disciplines in establishing and promoting a new concept of forest therapies. Thus, it promotes the use of natural resources and contributes to rural area development by means of new and attractive offer.

A list of all selected case studies with a short description and their main characteristics is provided in Table 1. Additional information is available in Appendix C.

**Table 1.** Description of selected social innovation cases (elaborated by the authors).

| | Name | Location of Case Studies from North to South | Description of the Social Innovation | Legal Registration Form |
|---|---|---|---|---|
| CS1 | Vojvodina House (Vojvodjanska kuća) | Stanišić village, Sombor municipality | A village women's association promoting self-sustaining economic initiatives. Focused on modernising the rural practices and experiences, e.g. by means of education in organic agriculture and production and agro-tourism. The goal is to empower women who are victims of violence. | Association of Citizens "Women's Association Udahnimo zivot" |
| CS2 | Forest Therapy (*Šumska terapija*) | Active in different rural regions | The association provides education for conducting "forest therapy" treatments, a spectrum of techniques or treatments for improving mental and/or physical health. Work is based on combining expert knowledge of several sectors (e.g. medicine, forestry, food) | Association of Citizens "Medical Spa Association" |
| CS3 | Garden of Sustainable Development (Avlija održivog razvoja) | Bogatić village, Šabac municipality | This initiative integrates social protection, agriculture, tourism and hospitality services. It aims to further the social integration and inclusion of people from socially vulnerable groups and to promote rural development in marginalised areas. | Association of Citizens "Caritas Sabac" |
| CS4 | "ForFriend" (ZaDruga) | Šabac municipality—more villages | A social enterprise designed to support small and medium-sized agricultural producers from rural areas. They harvest fruits and vegetables from their family orchards, which are dried and packed in high-value products. | Association of Citizens "Initiative for Development and Cooperation" |
| CS5 | Development Agriculture Fund Fenomena (DAFF) | Kraljevo municipality—more villages | Association of Citizens Fenomena established the "Development Agriculture Fund Fenomena" (DAFF), which operates as a "business angel" in support of integrated, sustainable agriculture in Serbia. | Association of Citizens "Fenomena" |
| CS6 | Rural Hub | Vrmdža village, Sokobanja municipality | Co-working space located in Vrmdza village is a community of creative individuals and an innovative organisation that aims to explore, build and connect its urban and rural knowledge in a sustainable way. Efficiently working on activating local populations for various activities for community development and wellbeing. | Association of Citizens "Centre for Education and Personal Development" |

**Table 1.** *Cont.*

|  | Name | Location of Case Studies from North to South | Description of the Social Innovation | Legal Registration Form |
|---|---|---|---|---|
| CS7 | First social agricultural cooperative (Prva poljoprivredna socijalna zadruga) | Kamenica village, Nis municipality | Engages young people who have land and resources for vegetable and fruit production. Jointly they produce value-added products, branded as "Art of flavours" enabling the producers to earn more and be better placed in the market. | Social cooperative, initiated by "Kamenica Local Development Association" |
| CS8 | Radanska ruža | Lebane municipality—more villages | A social enterprise securing employment for women from vulnerable groups, especially women with disabilities. Collaborating with local producers in partnerships to assure the availability of agricultural resources and then producing natural fruit-products based on traditional recipes. | non-profit limited liability company, initiated by "Women's association Ruza" |
| CS9 | Optimist | Bosilegrad municipality—more villages | A social enterprise producing non-wood forest products and employing poor and vulnerable groups. Working in partnerships with families for collecting and processing non-wood forest products and promoting these products to the broader public. | Association of Citizens "Optimist Bosilegrad" |

*4.2. Factors Supporting Social Innovation*

All case study representatives identified that a growing interest to engage in social innovation work represents the greatest potential for their work. They observe that some people are getting involved because of economic needs, but some are drawn in because of their attitudes, such as being pro-environmental, a desire to volunteer, or being inclined toward political activism. Many of them also feel the need to take personal responsibility for community development. Even though this interest was not as high at the beginning, it can be seen from interviews that there is growing solidarity around and trust in their work.

"My biggest investment and personal investment is in people ... we need to have patience, they (people) don't trust you upfront, first, they check it out in practice and if what they see makes sense to them then they are ready for change and for further learning." (CS6)

Many of the social innovation analysed for this paper benefited from the increased involvement of women and families, as well as some disadvantaged groups, which then resulted in them enjoying greater public acceptance and recognition.

"Family is the key to survival, in rural social innovation, because a man alone in the village cannot succeed" (CS6)

The most significant source of impetus in establishing and running the social innovation cases has been strong international donor support [63]. Direct funds from the European Union came from the EU PROGRESS programme, which is part of the EU Programme for Employment and Social Innovation (EaSI), European support to municipal development programmes (EU PRO), and the instrument for pre-accession assistance for rural development (IPARD). Support was also provided by national governments, e.g. from Switzerland (Swiss Pro programme) as well as donors from among UN organisations such as UN Women or development agencies such as the German Corporation for International Cooperation (GIZ), the Austrian Development Agency (ADA), the United States Agency for International Development (USAID); and international foundations such as the Rockefeller Brothers Fund, the Heinrich Böll Foundation, or organisations such as HELP-Germany, Caritas Italy, and Caritas Austria (Appendix C).

Along with foreign donor support, domestic banks now also offer special loans or grants to support social enterprises. One example of this comes from Erste Bank, that developed a loan programme called "Step by Step" which is backed by a European Union-funded guarantee under the auspices of the EU's EaSI Programme established by EU Regulation (No 182/2011. 1296/2013) of the European Parliament and of the Council [68,69]. A further example is that of UniCredit Bank's "Ideas for Better Tomorrow" programme that provides financial grants for projects with a clear social component. This programme is conducted in cooperation with the NGO Smart Kolektiv and the Ana and Vlade Divac National Foundation [70].

Active support is also provided by the very engaged non-governmental (NGO) sector at the national level. The most active in this regard is the Coalition for Development of a Solidarity Economy (KoRSE), which is an informal network of Serbian NGOs created in 2010. Members of the coalition are the TRAG Foundation, the European Movement in Serbia (EPuS), Group 484, the Initiative for Development and Cooperation (IDC) and the Smart Kolektiv. At the moment, the KoRSE coalition drafted the Law on Social Entrepreneurship in cooperation with GIZ. Representatives of the National Alliance for Local Economic Development (NALED), Citizens' Initiatives, Eurocontact, Group 484 and other interested representatives drawn from the civil sector all participated in the drafting process and the proposal has now been submitted to a Working Group of the Ministry of Labour, Employment, Veterans' and Social Affairs for discussion [71]. Outside of this coalition, the Social Economy Network of Serbia (SENS) has also been very active in promoting the work of social enterprises and innovations. Furthermore, national foundations (i.e. Trag, Delta, Ana and Vlade Divac) are also supporting social

innovations. All the aforementioned organisations supported our cases, either in terms of direct financing of their work or by providing advisory and educational support to increase knowledge capacities of the initiatives' representatives and users. The following quote highlights the importance of these contributions to one of the case studies we considered:

> "You cannot rely on state organisations, for them, it is important to end the conversation with you as soon as possible, so this assistance from the TRAG Foundation and the SMART Kolektiv was so very, very important." (CS9)

As regards public bodies it is important to emphasise the role of the Social Inclusion and Poverty Reduction Unit (SIPRU), which was formed by the Serbian government in 2009. Since 2018, it has operated within the Office of the Prime Minister. The mandate of SIPRU is to strengthen government capacities to develop and implement social inclusion policies based on good practices in Europe. However, it should be noted that this body is financed by the Swiss Confederation for a limited period of time. One of the major successes of the SIPRU team was to enable direct financial support to social enterprises through the IPA 2013 programme (Instrument for Pre-Accession Assistance) [72,73]. The initiative studied in CS5 was supported by this programme, but SIPRU was also important in other cases in terms of consulting and the promotion of results.

Improved access to technologies also contributed to the successful work of the analysed social innovations. e.g. by providing enhanced means for communication, exchange information and to access knowledge. Another supportive factor for all our case initiatives was the existence of valuable natural resources and favourable conditions, e.g. for growing marketable plants. Furthermore, social innovators recognised and benefitted from being able to meet new societal needs and demands, such as the growing demand for handmade, organic, and healthy products, and indeed, for more healthy lifestyles generally. These new demands proved to be supporting factors by providing opportunities that were helpful for all our case initiatives, by opening up markets for their products and providing impetus to innovate. In some cases, the demand for organic products has been so great that it has even allowed some initiatives to export to foreign markets.

### 4.3. Factors Hindering Social Innovation

Our analysis revealed numerous hindering factors as well. We grouped identified barriers according to the following aspects: local level policy-making, national policy-making, political influence, interest expression or representation, administration and bureaucracy, finances, social aspects, communication, coordination, education and skills, market, technological and finally infrastructure (Appendix B). In the following text we will elaborate on some of the most prominent hindering factors.

When it comes to "social" innovation, the issue of terminology is a widespread problem, as is evidenced that the terminological difficulties exist within many of the various barriers that we identified [73]. In the Serbian language the term "social" usually implies that something relates to social policy (welfare), which in the minds of many narrows down the meaning and then hinders application to the support mechanisms of social innovations which, unsurprisingly, in turn limits SI's potential. This view is also rooted in Serbia's history as the term "social" often related to "socially owned" enterprises in the socialist era. This creates reluctance on the part of people to engage in collective endeavours to some extent. Spear et al. [74] describe this as a problem which is a common issue in many former socialist countries. Thus, NGOs often employ terms such as "societal innovation" ("društvena inovacija") or "solidarity innovation" ("solidarne inovacije", related a solidarity economy) as these are perceived not be stigmatised with a negative connotation [63,73]. Such terms capture a broader spectrum of topics and issues that are societally relevant and should be addressed under the umbrella of "social innovations" (CS 1, 3, 4, 6, 7 and 9).

Considerable dissatisfaction of interviewees results from the inactivity of local municipalities. Our respondents point to a lack of interest and virtually non-existent support from local administrations, which is accounted for by both the disinterest of civil servants for the topic (lack of sensitivity) and the fact that social innovation is not placed on their agenda.

The inertia of local administration for not taking up new issues, resulting in them not properly targeting local needs, was stressed throughout our research. This is illustrated by the inefficient allocations within local budgets. Such budgetary allocations are usually not taken up, due to the fact that they finance activities which are of no interest to local communities.

"It would be much more beneficial if the local government would think about the interests of the local population" (CS5)

Respondents also pointed to a lack of capacities in local administration to deal with their issues.

" … some good legislation or initiatives from the higher level, people in local governments cannot follow because they do not have the capacity" (CS3)

The recent draft of the Law on Social Entrepreneurship (proposed by the by the Ministry of Labor, Employment, Veterans' and Social Affairs) is discouraging for all case representatives. They perceive that ministry in charge of the law will introduce a very narrow understanding of social entrepreneurship and innovation concepts, narrowing it down to the employment of vulnerable groups. The long and non-participatory process of drafting this law [71], together with the inactivity of the state in this field, indicates a lack of strategic and sustainable planning. This can be illustrated with following quote:

"There is a lack of a body who would essentially deal with this issue at some strategic level, not from government to government, but in a more permanent process, and so that the state sees the potential in it." (CS4)

According to many respondents' views, it would be better if this law is never enacted because of the constraints it may introduce. Furthermore, poor enforcement of laws is also generally seen as a problem.

"So you have laws, but you don't have a realisation ... you don't have the infrastructure that goes with that realisation, whether it is laws or by-laws ..." (CS6)

Some interviewees reported that they cannot rely on the state in some cases and that this insecure position sometimes threatens the existence of initiatives.

" … you have an absurd situation—a good, efficient business, beneficial to the people, comes to be at risk because the state simply does not fulfil its obligations... and you have no instrument to force them because you are at the bottom in that hierarchy." (CS3)

Regular changes in government add to the challenge as policy programmes change and so do state activities. Sudden and repeated changes make it difficult to rely on state organisations. In some interviews, differences between the current and previous governments were emphasised and respondents identified an increased distrust in civil society activities coupled with increased political pressure. In general, they saw the impacts of politics as a big threat rather than a boon in all societal spheres. As indicated by one of the respondents, everything at the local level is now extremely political. They reported that since they are not members of the ruling party it is forbidden for the local commission to approve funds to them, and they were told this was the reason.

Furthermore, respondents reported about corruption and illegal practices, both at the local and the national level and also about the misuse and a lack of transparency with regards to the expenditure

of public funds, such as abuses concerning funding non-governmental organisations. In this way these organisations are discouraged to apply anymore for funds.

The indecent offers to some of the case studies were as well pointed out. One respondent among our interviewees indicated that help for accessing funds is offered only against prior payment to persons involved in decision-making. Other reported that they were offered approval of organic certification without fulfilling prescribed criteria.

A lack of transparency concerning the provision of funds is noted as a challenge also when it comes to funding by donors. Some organisations think that there are no clear rules about who gets the funding and also report that some recipients are changing their focus depending on donors' preferences. This shrinks the pool of resources available for them but also results in a loss of trust.

> "There are some, not to mention now names, organisations that, between us, we call 'sects'
> … I am ashamed to know that there are organisations that for 10 years have gotten the same
> international donor money for every project … with absolutely no results behind them." (CS8)

The lack of financial resources is generally recognised as an issue where the greatest concerns involve meeting the costs of human resources, which are often not covered by donors. Thus, much of the work is necessarily done on a voluntary base, which is only possible because of the heartfelt enthusiasm and persistence of the people who are involved. The following quote typifies the concerns arising from the lack of funding for human resources:

> " … you have projects, but you do not have the resources for the people who need to
> implement the projects, which is, in my opinion, a great barrier to the development of the
> third sector in Serbia." (CS3)

Dependence on foreign donors creates a very unstable situation for all cases. They agree that ongoing financial support from the Serbian state would be beneficial. The current state practice of providing a one-time investment is not seen as providing a sustainable situation, especially not to local communities.

> "Giving one-time grants does not essentially lead to any further progress, neither of those
> supported farms nor of the community." (CS4)

Dealing with government, or project applications, administration and bureaucracy is reportedly very complicated, both when trying to obtain funds and also when fulfilling legal requirements in some cases, e.g. obtaining certificates, licences, etc. for organic produce.

Respondents also indicated a number of social aspects that were hindering factors, most notably all case studies reported that it was very hard to motivate people to become engaged in various projects.

> " … we faced deep distrust from the local population, the broader picture was not clear
> to them … to do something together, to sell and then distribute money afterwards was
> somehow not clear to them … They have logic 'we give you the goods, you give us the
> money'" (CS4)

Moreover, our respondents noted increasing apathy among people in the last few years, primarily explained by the unstable socio-political conditions in the country. The result of this was seen in the fluctuation of interested, involved people and the loss of interest after some time which was also at times connected to excessively high initial expectations not being met (e.g.in terms of economic benefits). Challenges in sustaining a positive community spirit were reported as the benefits gained are not derived immediately and frequently need considerable time to bear fruit. Human nature being what it is, it was also reported that it was difficult for individuals in needy sections of rural society to rate community interests over direct, personal interests.

> " . . . it is important that people understand that we do this not only for the association, just for a group of women, but that it is for the wider community . . . " (CS1)

The especially vulnerable position of women in rural areas was reported in many cases as a barrier. According to our respondents, the status of women is not effactually recognised. Therefore, women are quite often seen as the most important target groups of social innovation activities.

> "Women are 'another' category. I have women in the association who had some kind of support . . . and if I remember, they faced many inconveniences, not just from family but their wider surroundings . . . You can empower women by pointing out that they are the ones who can earn, but it is important to put them always in the context of family." (CS6)

Respondents also reported about the inactivity of people in rural areas, who often rely on state support and work. Rural areas also lack effective leaders and, even if a leading figure is found, a whole initiative or business can be placed at risk if it becomes overly dependent on the one person.

A lack of communication on all levels, as well as coordination among different actors and political levels combined with the lack of education and skills (in different fields of expertise) were also reported as hindering factors.

In relation to more contextual factors, a lack of infrastructure in rural areas is still a huge problem [69], even though with the passage of time some improvements can be observed. Market access and difficulties to become competitive in the market are seen as further barriers to the success of social innovations. This is illustrated with following quote:

> "However important that social character is, it is important that the product is affordable . . . and on the other hand, that (social character) may be our competitive advantage at the market." (CS4)

## 5. Discussion

> "Until the supporting system for our business is developed, the problem of our survival is enormous, and we are really making a superhuman effort just to survive." (CS8)

Similar to other transition economies [60], social innovation and social entrepreneurship are a relatively new phenomenon in Serbia when compared to their profile in developed countries [73,75]. Looking at social innovations from an institutional void perspective, and by understanding the situations in the analysed cases by identifying supporting and hindering factors, we are able to ascertain a number of institutional voids in the current institutional setting in Serbia and relate them to the institutional factors that help to overcome the deficiencies resulting from these voids (Table 2).

The lack of regulatory frameworks or strategies on social innovations and entrepreneurship has been identified as a formal institutional void. Social innovation organisations have to navigate between existing regulations and/or tailor their business models to fit existing rules while accepting adverse effects for their businesses. The introduction of a "social cooperative" category within the Law on Cooperatives (2015) is seen as an institutional measure that partly fills this gap [59,67] and was certainly supportive for the CS7 business model, for example. According to this law, social cooperatives undertake various activities to promote the social, economic, or other related needs of vulnerable social groups. Social cooperatives are obliged to invest at least half of their profits into the improvement and realisation of a set of social objectives which are explicitly contained within each cooperative's statute [59].

However, all our case studies stressed that while they are able to work within the existing regulatory system, they all expressed a need for better-tailored policies when it comes to social innovation, either in the form of a law or a specific national level strategy. In this regard, we have to be mindful that our cases represent a sample that features a positive bias as we did not research ideas and initiatives that had failed to successfully establish themselves and did not become operational

for a significant period of time. Indeed, it may well be that a number of initiatives failed shortly after being established because of this lack of an appropriate regulatory framework. Addressing this situation has proven to be problematic in Serbia, as is evidenced by the already 10-year long process of developing the Law on Social Entrepreneurship in which the ministry has proposed three different drafts. None of these drafts met the expectations of the concerned social enterprises and NGOs. As recently as September 2019, the KoRSE coalition, in cooperation with GIZ, drafted a Law on Social Entrepreneurship which was in line with proposals received from civil society actors. This draft law was submitted to the ministry and is now under discussion at the political level [71], which will naturally further extend the period of regulatory inadequacy.

**Table 2.** Indicated institutional voids in the case studies involving social innovations in rural areas of Serbia and related supporting factors to overcome voids (elaborated by the authors).

| Identified Voids in Case Studies and Supporting Factors to Overcome Voids | | | |
|---|---|---|---|
| **Formal Institutional Voids** | **Supporting Factors to Overcome Formal Institutional Void** | **Informal Institutional Voids** | **Supporting Factors to Overcome Informal Institutional Void** |
| Lack of and poorly enforced regulations for social innovations | Law on cooperatives (2015)—"social cooperative" Draft Law on Social Entrepreneurship proposed by NGOs | Traditional norms and values constrain more productive resource use | Incentives to sell to export markets assisted by certification programmes |
| Lack of financial mechanisms for supporting social innovations | Specific funding lines by foreign donors Specific financing mechanisms by the domestic banking sector and foundations | Weak position of rural women in the patriarchal system | Programmes for involving and empowering women Potentially gender-responsible budgeting |
| Absence of institutionalised intermediary organisatons | SIPRU unit and the KoRSE coalition could assist the government in their activities | Some accepted level of corruption/acceptance of poltical elites misuse of power for self-enrichment | no specific counter-factor identified |
| Lack of cooperation mechanisms between state organisations, and between state and non-state actors | NGOs formed a coalition to coordinate activities (KoRSE Coalition) Social enterprises are joining an association of social agro-businesses | Lack of informally institutionalised coordinative mechanisms | |
| Inadequate (institutionalised) provision of specialised information | KoRSE Coalition, SENS network and SIPRU serve as platforms to support information exchange | Lack of trust and solidarity in society | |
| Lack of formal educational institutions | NGOs providing training and mentoring | Apathy within parts of society | |
| Non-participative procedures by governmental bodies | no specific counter-factor identified | | |
| Incongruence of national and local policy-making and implementation | | | |
| Insecure contracts with state | | | |
| Weak position of civil society | | | |

The critique of the existing legal framework as raised by respondents strongly highlights their discontent with being treated as any other profit-oriented business and the lack of state financial support mechanisms specifically assisting social innovations and enterprises. This void has been filled by active financial support provided by foreign donors, the domestic banking sector as well as financial support by private domestic foundations (see supporting factors). Case five features a small-scale example of how the establishment of a social innovation itself, i.e. the establishment of the "Development Agriculture Fund Fenoemena" (DAFF) aims to overcome the lack of institutionalised funding opportunities by operating as a "business angel" in support of integrated, sustainable agriculture.

Compounding all of the above problems, even where workable rules exist poor enforcement is often perceived as facilitating opportunistic behaviour. Local administrations, for example, are perceived to foster those rules which benefit specific societal groups, i.e. their clientele. This may be accompanied by a lack of transparency and participation in procedures, such as public budgeting and spending, policy formulation, and policy implementation at both the national and the local level. Furthermore, national and local policy-making and implementation are not always congruent, leading to gaps, missing rules, or even contradictory rules being applied. As a result, organisations active in social innovation cannot fully rely on state structures, even when it comes to formal contracts. Similar voids were reported in the study on Ukrainian entrepreneurship development in a transition context, where the government has yet to fully implement an effective institutional framework for productive entrepreneurship [76].

A general lack of provision of relevant information on social innovations has been noted throughout Serbia. Many of the social innovation initiatives are completely reliant on personal contacts and knowledge. Additionally, a lack of education and educational support seems evident, resulting in labour markets having to draw on an unskilled and ill-prepared workforce. To improve the knowledge and skill base for social innovations' workers would prove to be prohibitively expensive resource-consuming activities that the individual social innovation organisations would have to bear themselves. This was also confirmed in a study on the institutional voids related to the business environment in Serbia and Turkey [49]. Thus far these gaps were partially addressed by NGOs which provide training, advisory services and mentoring activities with the SIPRU unit also working to some extent to support social innovations in this regards.

A further significant void was found in the absence of institutionalised intermediary bodies, namely organisations which should be dedicated to the coordination and support of social innovation initiatives. The literature on institutional voids already emphasises these kind of deficiencies, which "occur when specialised intermediaries are absent" [50] (p. 184). Social enterprises or civil society organisations which advocate improved working conditions for social entrepreneurship and innovation may take over this role in the future [32]. The SIPRU unit and the KoRSE coalition could provide practical assistance to the government via their activities in support of social innovators. The extent of these roles and activities has been rather limited (with sporadic activities). However, civil society organisations are in weak positions in Serbia. Their activities are distrusted by the state. Such an environment is not supportive for "bottom-up" initiatives, as was also confirmed by the report of BTI [77] which noted that the interests of civil society organisations are not highly regarded in public policy discussions, both nationally and locally. Civil society organisations influence public policymaking by an individual or joint coalition initiatives, e.g. in areas such as the EU accession process with regards to issues of human rights, youth unemployment, environmental or security issues. The stigmatisation of civil society organisations started in the 1990s in Serbia and has gained considerable momentum in the last couple of years. They are often described as "foreign mercenaries" and "domestic betrayers," not only by some political parties, far-right extremist groups, and certain tabloid media outlets but also by representatives of the government.

The lack of institutionalised cooperation between state organisations, as well as between the state and other various private sector actors was also identified as a formal void, a problem which may partly stem from the strong sectoral fragmentation of the public administration [23,30]. However social innovations, as with innovations in general, are dependent on external knowledge and competences.

The provision of such competences could be facilitated by the formation of cluster structures, i.e. geographic agglomerations of companies, suppliers, service providers, and associated institutions in a particular field, linked by externalities and complementarities of various types [78]. Cluster organisations would potentially support networking among regional enterprises, facilitate knowledge exchange and cooperation, improve access to investments, to subsidies, to training and to research and development services. Such structures could provide invaluable support to innovation system functionality [49,79]. In fact, the KoRSE coalition advocates for such multi-sectoral cooperation in the social enterprise sector. Some of the representatives of the social enterprises analysed in this paper also see great potential in joining forces, e.g. in an association of the social agro-businesses, a process which is still in its initial phase (CS3, 9).

Our data also hint at a high number of informal institutional voids, which also harmonises with some other scholars' results that indicate the importance of normative and cultural voids in certain contexts [6]. The study in Bulgaria identifies 'institutional asymmetry' between formal and informal institutions which hampers the development of economically and socially productive entrepreneurship. The authors claim that despite reforms to formal institutions the asymmetry persists as a result of irregularities within informal institutions, such as entrepreneurs engaging in informal and corrupt activities [80].

From the examination of our cases, we identified various informal institutional voids. Some are related to traditional resource use where it is difficult for rural people to change existing practices used in agricultural production and to engage in local community activities. Societal needs for organic products for example, pushed some of the initiatives researched in our case studies to undertake organic agriculture according to prescribed standards. Thus, the introduction of the organic farm products certification process induced real world changes in some local agricultural practices.

Tradition and a (still) dominant patriarchal system create another informal institutional void that in particular results in a weak position for women, especially in rural areas. There is some persistent prejudice that women's work only concerns the household and not agricultural production. In many cases, this becomes even more challenging as women are usually not the legal owners of agricultural holdings. Thus, they are in an inferior position to their husbands or other males in their families. In some of our cases, their engagement in social innovation initiatives faced consternation in or even mockery from their communities and families which took much time and energy to overcome. This void is being addressed by various programmes (incl. the rules and goals of funding by donors) that are aimed at including and empowering women, but also other vulnerable groups and minorities whilst simultaneously raising awareness of the valuable roles of these groups. At the national level, a few regulations were enacted which have applicability to overcome this gap, including the new National Strategy for Gender Equality 2016-2020 and an action plan for 2016-2018 [77], and rules for gender responsive budgeting adopted in 2015 [73]. However, a European Union Report noted a serious delay in passing these regulations [81] and also went on to note that the institutionalisation of the coordination body for gender equality still needs to be clarified and an efficient institutional set-up with adequate resources needs to be ensured. The report furthermore stresses that older, rural and Roma women as well as women with disabilities continue to be among the most discriminated against groups in Serbian society.

Furthermore, the manipulative use of the power of public administration, both at the national and municipal level, entails informal voids. Public resources are misallocated and community development is steered in wayward directions not always corresponding to the needs of rural populations. Reported cases of corruption also demonstrate the misuse of this power. Indeed, the European Union concluded that corruption is prevalent in many areas throughout Serbia and remains an issue of concern [81]. All this creates a serious lack of trust between communities and social innovators that limits individuals' willingness to engage in relational and investment activities. Another very relevant void is manifested in the growing general apathy of people, often seen as a result of many years of socio-economic crisis in Serbia, but also due to value systems which favour political nepotism.

Quite a number of the institutional voids relevant for social innovation, as discussed above, are also relevant for the broader business community in Serbia. There are many burdensome procedures and overlapping authorities, as well as a high incidence of corruption among state officials and bureaucrats. The legal framework is partly inconsistent and prone to unexpected and significant changes, all of which is detrimental to any entrepreneurial endeavour [77].

## 6. Conclusions

We have drawn on institutional theory to examine how formal and informal institutions, both directly and indirectly, influence the development of social innovations in rural areas in Serbia. The influence of institutional voids is particularly pronounced in unstable institutional environments, often found in countries such as Serbia that are in transition. During a transition process, countries find themselves experiencing constant changes in their institutional environment. This poses extraordinary challenges for companies, producers, and civil society organisations to adapt to and function under these conditions. Policymakers usually have a strong focus on the state and further development of the formal institutional environment, hence, adapting and adopting laws and regulations to suit the new situations and desired political and economic models. The valuable role of informal institutions is often underestimated, neglected, or even completely ignored and thus giving rise to institutional voids [49].

Overall, our case studies show that numerous institutional voids exist in Serbia when it comes to the development of social innovations and enterprises. All the case representatives reported insufficient supporting activities and understanding of the concept by both national and local authorities, coupled with inadequate or nonexistent regulatory and financial mechanisms, a lack of coordinating bodies, and dysfunctional communication channels and educational offers, all of which need to be improved to effectively support social innovation activities. These formal institutional voids are furthermore accompanied by informal voids such as norms rooted in traditional societal beliefs which constrain the productive use of resources and continue to neglect and discriminate against certain groups of society.

All the analysed social innovations in Serbia operate successfully to some extent, but under very unfavourable conditions. Currently, they have to contend with a somewhat hostile environment, given the existing regulatory system and societal context. All case study initiatives are highly dependent on external financing by donors, thus they operate under rather tenuous circumstances and struggle to sustain their operations. More stable and innovative financing mechanisms are needed. Nevertheless, a high level of interest and activity by national non-governmental organisations is very important for the proper functioning of the analysed social innovation initiatives. Together they work to advocate a broader understanding of the societal value of social innovation and entrepreneurship and to ensure that adequate national legislation is passed that supports such endeavours. With regards to some of these formal and informal voids that became manifest through our case studies, we identify related supporting factors that—at least to some extent - are helpful to and have some potential to overcome obstacles created by the various institutional voids. Even though, the "sector" of social innovation is still in an infantile state, some improvements were achieved over the course of the last decade.

All our considered case studies are engaged in strong socially-oriented activities, with some focused on social entrepreneurship activities because of their subsistence-oriented aims, namely satisfying 'survival' needs and the need to reach a financially sustainable position. These tend to then gradually transform into growth-oriented aims, which lead to productive benefits across product markets and create employment opportunities. Other case studies we examined follow quite idealistic and life-style oriented aims, pursuing innovative solutions which are less oriented to market-based results. Generally, they all succeed in offering new options and approaches which serve to motivate and involve rural populations and build trust among community members. Our case studies provided a comprehensive view of the issues that relate to institutional challenges for social innovations in rural areas in Serbia. Obviously, this is a fertile area for future research, with high societal relevance. Avenues for improvement in this field are to be further researched. Our intention with this piece was

to bring existing challenges to the forefront and to lay a foundation that will serve to stimulate not only future research, but also policymakers, development agencies and other interested actors to strengthen their support for social innovations as a means to sustainably develop all rural areas that could benefit from such innovation, not just those in Serbia

**Author Contributions:** I.Ž. undertook the work in overall research design and conceptual framing, case selection, data collection, data analysis, writing the paper and lead the work on the revsion of the paper, K.H. and A.L. contributed in conceptual framing, paper writing and improving the paper based on external reviews.

**Funding:** The research was conducted within the European Union's Horizon 2020 research and innovation program under Grant Agreement No 677622 (SIMRA). The open access publication was supported by BOKU Vienna Open Access Publishing Fund.

**Acknowledgments:** We are grateful to all our interview partners for spending their time and efforts on answering our questions. We thank three external reviewers who helped to significantly improve this paper.

**Conflicts of Interest:** The authors declare no conflict of interest.

## Appendix A

**Table A1.** Supporting factors for analysed social innovations (elaborated by the authors).

| Supporting Factors | Codes | Groundedness |
|---|---|---|
| Social aspects | High enthusiasm, persistence and volunteering of involved people is key | 8 |
| | Built trust | 6 |
| | Solidarity is important | 4 |
| | Women as potential for rural areas | 2 |
| | Family as key to sustainability | 1 |
| | Idealistic approach, without any rational approach | 1 |
| | Interested users/target group | 1 |
| | Personal attachment of employees to company | 1 |
| | People's sensitivity to social problems | 1 |
| Policy aspects | Sufficiently broad legislative legal environment that can be utilised | 5 |
| | New regulation on the production of fruits, vegetables, dairy products | 2 |
| | Law of Professional Rehabilitation provides financing for PwD | 1 |
| | The Law on Cooperatives provides for the possibility for forming social cooperative | 1 |
| Governance aspects | Networking of social enterprises and initiatives | 4 |
| | Bottom-up initiatives are important | 1 |
| | Cooperation with high schools | 1 |
| | Cooperation with NGOs that work in this sector | 1 |
| | Personal connections are important | 1 |
| Donor aspects | International donors and advice important for starting | 8 |
| | Donations came also from private sector | 1 |
| | Philanthropic investing | 1 |
| Communication aspects | Good examples stimulated others to join and increase visibility | 5 |
| | Public acceptance and recognition | 3 |
| | Advocacy role is important | 2 |
| | Mentoring is important | 1 |
| | Relaying of friends and personal contacts | 1 |
| | Support of media is important | 1 |
| Incentives | Obtaining certificate for picking of wild products | 1 |
| | Obtaining certificate for business plan preparation | 1 |
| | Offering higher prices for raw material | 1 |
| Knowledge/Skills aspects | Importance of knowledge transfer from practice/NGOs | 6 |
| | Education possibilities of local people | 3 |
| | Mutual learning | 2 |
| | Skilled team for management | 1 |
| Technology aspects | Availability of technology | 2 |
| | Internet connection | 1 |
| Market aspects | Importance of good branding | 2 |
| | Placing product in the right market | 2 |
| | Geographic origin | 1 |
| | Territorial branding | 1 |
| New needs of society | Adapting traditional products to modern needs | 1 |
| | Importance of ecology issues | 1 |
| | Importance of healthy living style | 1 |
| | Interest in handmade, organic, healthy products | 1 |
| | Interest in organic products | 1 |
| | Forest bathing is a leading global trend | 1 |

# Appendix B

**Table A2.** Hindering factors for analysed social innovations (elaborated by the authors).

| Hindering Factors | Codes | Groundedness |
|---|---|---|
| Local level policy-making | Lack of interest of local administration | 11 |
| | No support from local administration | 5 |
| | Inertia of administration | 5 |
| | Inefficient budget spending | 4 |
| | Corruption | 4 |
| | Contradicting information and advice from local administration | 3 |
| | Contradictions with national strategies exist at local level | 2 |
| | Inspection is weak | 2 |
| | Local needs are not addressed by local administration | 2 |
| | Non participative decision making at local level | 2 |
| | Lack of capacities in local administration | 2 |
| | Communication with local administration is built on personal connections | 1 |
| | Fear to confront to the local administration | 1 |
| | Local administration equalise rural development and agricultural development | 1 |
| | Local administration reduce funding | 1 |
| | Local government in not reliable partner (as cofounding partner) | 1 |
| | Not functioning local administration | 1 |
| | Social responsibility is lacking in the local governments and employed people there | 1 |
| National level policy-making | Current draft Law on Social Entrepreneurship is discouraging | 13 |
| | Lack of strategic and sustainable planning | 8 |
| | Weak enforcement of law | 8 |
| | State does not recognise the potential of social entrepreneurship | 7 |
| | Hard to rely on state organisations | 6 |
| | Narrow understanding of social entrepreneurship and innovation concepts by state | 6 |
| | Inertia of administration | 5 |
| | Corruption | 4 |
| | Challenge of top down governing | 3 |
| | Changing government structures | 3 |
| | Distrust in NGO activities from state | 2 |
| | Lack of bylaws, regulations and measures | 2 |
| | Laws are not targeting small producers sufficiently | 2 |
| | Law on Associations limits opportunities for using state funds, or taking loans | 1 |
| | Low awareness of policy-makers | 1 |
| | Lack of regulations and financing mechanisms to support organic production | 1 |
| | Rural policy, after 2000, was without concrete aims | 1 |
| | State insists on incorporating social enterprises into the Companies Act | 1 |
| | Unequal support of Ministries to Vojvodina and the rest of the country | 1 |
| | We fitted our model to the existing regulations of the state | 1 |
| Political influence | Politically favorable organisations are supported | 3 |
| | Overarching problem is the impact of politics in all spheres | 1 |
| | Some civil servants installed politically | 1 |
| Interest expression or representation | Terminology issue of social entrepreneurship/social innovation | 5 |
| | Issue is addressed just by NGO sector | 1 |
| Administration and bureaucracy | Administration and bureaucracy is complicated | 9 |
| | Bureaucracy is very complex for organic production | 4 |
| Financial aspects | Lack of transparency in providing funds | 7 |
| | Lack of financing (in general) | 6 |
| | Lack of financing for human resources | 6 |
| | Lack of financing from the state | 4 |
| | Challenge of fitting donor's funds to various organisational forms | 3 |
| | Change of the donors focus is challenging | 2 |
| | Funds comes mostly from donors and foreign funds | 2 |
| | One-time investments are not profitable enough | 2 |
| | Misuse of financial resources | 3 |
| | Risk funding for donors | 2 |
| | Calls for funding do not relate to real needs | 1 |
| | Challenge to address high number of very small plot holders with financing mechanisms | 1 |
| | Costs for going on market is same for us and big companies | 1 |
| | High costs for licensing | 1 |
| | High personal financial investments | 1 |
| | It is hard to obtain finances for scaling | 1 |
| | Private businesses are more open for one-time support | 1 |

**Table A2.** *Cont.*

| Hindering Factors | Codes | Groundedness |
| --- | --- | --- |
| Social aspects | Hard to motivate people to join | 10 |
| | Status of women is not effectually recognised | 9 |
| | Fluctuation of interested people is challenging | 5 |
| | Hard to change existing practices | 3 |
| | Hard to perceive community interest over direct/personal interest | 2 |
| | Hard to rely on self-organisation and cooperation of community members | 2 |
| | High expectations of local people when they engage in social innovation | 2 |
| | Loss interest after some time | 2 |
| | High voluntary involvement | 2 |
| | Lack of leaders | 2 |
| | Apathy of people | 1 |
| | Challenge to sustain community spirit | 1 |
| | Culture of sanctioning those who make mistakes | 1 |
| | Ethics are of a low level | 1 |
| | Inactivity of people in rural areas-relying on state support | 1 |
| | Low awareness of the potential of resources readily available | 1 |
| | Risk if whole process depend on one person | 1 |
| | Skepticism in the potential of improvement at macro level | 1 |
| Communication aspects | Lack of communication | 4 |
| | Lack of information for rural people | 2 |
| | Need to have an intermediary actor who would support communication | 1 |
| Coordination aspects | Not willing to cooperate with state under current conditions | 2 |
| | Not satisfied with functioning of this public private partnership | 1 |
| | Private business are not interested in partnering with NGOs | 1 |
| | Superficial cooperation with the local government | 1 |
| Education and skills aspects | Lack of education of people living in rural areas | 6 |
| | Lack of human resources | 3 |
| | Lack of experiences | 2 |
| | Lack of knowledge on business functioning | 1 |
| | Lack of organisational skills | 1 |
| | Lack of skills for project writing | 1 |
| | Lack of willingness to learn new things | 1 |
| Market aspects | Small producers cannot be concurrent on the market | 2 |
| | Challenge of market valuation | 1 |
| | No potential for mass production | 1 |
| Technological aspects | Challenge of crating adequate technological process | 1 |
| | Small parcel cannot be productive | 1 |
| Infrastructure aspects | Lack of infrastructure | 5 |

## Appendix C

**Table A3.** Details of case studies (elaborated by the authors).

| | Case Study Name (with Name in Original Language) | Target Users | Cooperating Organisations | | |
|---|---|---|---|---|---|
| | | | International | National | |
| | | | UN, Funds, NGOs | NGOs, Funds, Banks | Public Organisations |
| CS1 | Vojvodina House (*Vojvodjanska kuća*) | Women who are victims of violence, unemployed women | UN women, Heinrich Böll Foundation, | fondB92, Delta foundation, Ecumenical women´s initiative | Agricultural Expert Service Sombor |
| CS2 | Forest Therapy (Šumska terapija) | Urban and rural population in general | Cross-border international projects | Private companies | Faculty of Forestry, High School of Health Professional Studies |
| CS3 | Garden of Sustainable Development (Avlija održivog razvoja) | Persons with special needs, handicapped, poor and socially disadvantaged, former prisoners or addicted, Roma and other minorities, elderly, young people | IPARD, GIZ, Caritas Austria, Caritas Italy | TRAG Foundation, European Movement, SMART kolektiv, SENS, Erste Bank | Local municipality (financing care services) |
| CS4 | "ForFriend" (ZaDruga) | Small and medium-sized agricultural producers, households | ASB Austria, USAID | Coalition for Solidarity Economy, Design Taste Center | Municipality of Šabac |
| CS5 | Development Agriculture Fund Fenomena (DAFF) | Small and medium-sized agricultural producers, households | SWISS Pro, GIZ, Rockefeller Brothers Fund, UN Women | Slow food network, SOS Children Villages Serbia | SIPRU, Municipality of Arilje and Kraljevo, Regional Development Agency Zlatibor, Agricultural Chemistry School, National Employment Service |
| CS6 | Rural Hub | Local population of Vrmdža village | GIZ | European Movement, Kamenica Local Development Association | Municipality of Sokobanja, SIPRU |
| CS7 | First social agricultural cooperative (Prva poljoprivredna socijalna zadruga) | Unemployed young people in the hard to employ category | GIZ, Rockefeller Brothers Fund | Delta foundation, Erste Bank | Ministry of Youth and Sports, Cooperative Union of Serbia |
| CS8 | Radanska ruža | Women belonging to vulnerable groups | EU Progress, | Caritas, Erste bank | Municipality Lebane (not succesful cooperation) |
| CS9 | Optimist | Women belonging to vulnerable groups, Roma families, young people | EU Progress, SWISS Pro, ADA, Rockefeller Brothers Fund | TRAG foundation, SMART kolektiv, Delta foundation, Erste Bank | Municipality Bosilegrad (superficial cooperation) |

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
