# Peer review of "Social Innovation to Sustain Rural Communities: Overcoming Institutional Challenges in Serbia"

_sustainability, doi:10.3390/su11247248_

Round 1

Reviewer 1 Report

MAJOR COMMENTS:

My main comment regards the second research question: “Are Social Innovation initiatives abloe to take advantage of supportive factors to overcome hindering factors?”. As it is formulated, the reader expects some sort of comparison between the intensity the hindering factors have on each initiative and how those effects are counterweighted by the supportive factors. Yet, in the text, the authors merely present one the one side the supportive and on the other side the constraining factors, without any comparison or assessment of how the first “overcome” the second. Therefore, please revise the text accordingly, either the results section or the formulation of the research question.

In addition, it remains obscure how the identified factors actually affect the emergence and implementation of the Social Innovation cases analysed in terms of agency of the entrepreneurs/innovative actors. I suggest the authors to benefit from the Planned Behaviour perspective or similar approach, where the (positive/negative) interaction between the identified elements with the social innovation idea and activities is reflected.

MINOR COMMENTS

Line numbering disappear after Table 2 –for a next round of revision, adding the lines will facilitate the reference to specific sections.

As a general comment, it seems that the authors employ “institutional void” as synonym for “hindering factors”. If that is the case, please, make it more explicit; otherwise, a clarification in the use of the terminology is needed.

In addition, in the text the term “institutions” is sometimes used as equivalent to “rules of the game”, being those formal or informal (as defined in L120-123), but the term is often also used as synonym of “public administration” (e.g. L128). Owing to the importance of the term along the text, please revise the text accordingly.

When presenting the concept of institutional void (Introduction), I wonder where the private/market instruments fall (certification, standards, etc), as they are not regulations but are no informal institutions neither.

There the are some logic jumps in the text, insofar as some sentences are disconnected from the immediate ones (e.g. L167, L320-321, paragraph before “hindering institutional factors”). Please, consider revising those paragraphs to make the reading smoother and more coherent.

The manuscript refers to "sustainable rural communities" but the analysis refers chiefly to rural communities. So, could you please elaborate on the "sustainable" adjective?

L215: weird sentence

How does the terminology of “social” affect the functioning of the analysed initiatives? (second paragraph of hindering factors). Why one term is perceived as more appropriate?

Table 1 is very similar to Table 3. Given that the added value of this paper falls in Table 3, and the contents of Table 1 are explained in the text, I recommend the deletion of the first table and rather employ the space for further clarifications.

The conclusions miss a reference on the informal institutional findings.

Author Response

RESPONSES TO THE COMMENTS of REVIEWER 3

Manuscript ID: sustainability-655137

Title: Social innovation for sustainable rural communities: overcoming institutional challenges in Serbia

Article type: Research Paper

Authors want to thank the third reviewer for the time and effort spent on the paper. The comments and suggestions are very valuable and helped to increase the quality of our paper significantly. We considered and addressed all of them, as described in the following table. We hope that the revised manuscript meets all the requirements and expectations of the both reviewer and the editor.

All new and changed text-parts, of the now once more revised manuscript were sent to a professional native speaker proof-reading (see the attached conformation of the proof-reading service).

All parts of the text which were added or change based on reviewer comments are in the track-change mode in the manuscript.

REVIEWERS 1 COMMENTS

answer

MAJOR COMMENTS:

My main comment regards the second research question: “Are Social Innovation initiatives able to take advantage of supportive factors to overcome hindering factors?”. As it is formulated, the reader expects some sort of comparison between the intensity the hindering factors have on each initiative and how those effects are counterweighted by the supportive factors. Yet, in the text, the authors merely present one the one side the supportive and on the other side the constraining factors, without any comparison or assessment of how the first “overcome” the second. Therefore, please revise the text accordingly, either the results section or the formulation of the research question.

We are very thankful to this comment. In fact, going back to our data, we were able to clearly relate some voids and supportive factors for social innovation, and thereby to sharpen and improve the analytical part of our paper. These relations were not directly addressed in the previous version of the manuscript.

The second research question is as well reworded: “Which supportive factors are helping to overcome identified institutional voids?” (L119)

The revised version addresses these relations in the discussion section, describing the relation of supporting factors to institutional voids. Some of them are in place and fill a void to certain extent, while other supporting factors are at the stage of development/discussion but have the potential to fill gaps.

All these changes are in track-change mode in revised manuscript (L671-677, 685-692, 697-702, 723-725, 732-734, 756-759, 772-775, 783-787)

Furthermore, we also included a new table 2 (L666) which presents institutional voids and corresponding supporting factors.

In addition, it remains obscure how the identified factors actually affect the emergence and implementation of the Social Innovation cases analysed in terms of agency of the entrepreneurs/innovative actors. I suggest the authors to benefit from the Planned Behaviour perspective or similar approach, where the (positive/negative) interaction between the identified elements with the social innovation idea and activities is reflected.

We thank you for this comment, and see valuable potential in use of Ajzen (1993) Theory of Planned Behaviour, and similar concepts and analytical frameworks.

However, the focus of this manuscript was and is on the institutional setup for developing social innovations. Interview guideline was developed accordingly, also the literature review and the document analysis was guided by this research focus. Therefore, we don’t have sufficient data to seriously apply frameworks such as the TPB systematically and fruitfully yet. But we consider this for further cases studies and studies with enlarged number of cases, maybe with more structures, even quantitative survey for investigating relations between hindering and supportive factors and e.g. attitudes, social norms, perceived behavioural control and behavioural intentions of social innovation actors/entrepreneurs, ideally also including failed cases.

MINOR COMMENTS:

Line numbering disappear after Table 2 –for a next round of revision, adding the lines will facilitate the reference to specific sections.

done

As a general comment, it seems that the authors employ “institutional void” as synonym for “hindering factors”. If that is the case, please, make it more explicit; otherwise, a clarification in the use of the terminology is needed.

Thank you for this comment.

Our intention was not to use “institutional void” as synonym to “hindering factor”. Actually, hindering factors are much broader in scope, and are not necessarily and not all directly related to institutional barriers. However, since the case of Serbia is rather rich in hindering factors and many related to the institutional setup, we found many such connected voids.

Clarification on this is given in the Method section (“In this study the term “hindering factors” encompasses a broad scope of factors that are or create barriers for social innovations. However, it must be noted that such hindering factors are not necessarily manifested as or even result from an institutional void. ” L216-219)

For example, one barrier for social innovators is weak road infrastructure and remoteness of these areas, but that is not institutional void. On the other side, a lack of formal rules, or state budgeting priorities and line that allow financial support is seen as void.

In addition, in the text the term “institutions” is sometimes used as equivalent to “rules of the game”, being those formal or informal (as defined in L120-123), but the term is often also used as synonym of “public administration” (e.g. L128). Owing to the importance of the term along the text, please revise the text accordingly.

Thank you for this comment. We carefully checked our text for the possible misuse of the terms.

We use term INSTITUTIONS as equivalent to “rules” or “rules of the game”, both: 1) formal institutions, i.e. (e.g. regulations, formalised programmes and formulated policies) AND 2) informal institutions (e.g. unwritten norms of behaviour, values, beliefs and cultural practices) (cf. North 1991); We use the term ORGANISATIONS (as “players of the game”) in terms of formal structures with explicit purposes that are consciously created.

In the theoretical background section, we provide explanation on what we understand by institutions more in detail (lines 142-156).

When presenting the concept of institutional void (Introduction), I wonder where the private/market instruments fall (certification, standards, etc), as they are not regulations but are no informal institutions neither.

We did not elaborate on non-state governance or non-state governance instruments such as non-state certification systems and standards in the section where we introduce the concept of institutional voids. However, we understand (private body) certification systems (e.g the system of the Forest Steward Ship Council, FSC) as non-state market driven governance approaches that apply formal rules, usually at a voluntary basis, but formal soon as e.g. a company applies to a certification body (i.e. the rules and procedures to gain certification and to use certificates are highly formalised for contract partners).

There are some logic jumps in the text, insofar as some sentences are disconnected from the immediate ones (e.g. L167, L320-321, paragraph before “hindering institutional factors”). Please, consider revising those paragraphs to make the reading smoother and more coherent.

We revised all instances that we found in this regards, and all paper was externally proof read professionally. Changes are now in track change in lines 193, 351, 482-491.

The manuscript refers to "sustainable rural communities" but the analysis refers chiefly to rural communities. So, could you please elaborate on the "sustainable" adjective?

We thank you for this comment, and agree that “sustainable” is not appropriate here. We did not realize this mistake, when we simply used the term lending from literature on social innovations which emphasises the potential of social innovations to foster sustainable development in rural areas. Our paper deals with aspects of sustaining rural communities in Serbia; we thus changed our title (“Social innovation to sustain rural communities: overcoming institutional challenges in Serbia”) and all the text accordingly; see lines 112, 138, 841)

L215: weird sentence

We reformulated this sentence. Now it is as follow “Through an iterative process, initial codes were grouped into more focused and substantive categories of supportive and hindering factors, which are of relevance to the particular case studies (see Appendix A and B) and these codes were then related to concepts of formal and informal institutional voids.” (line237-240)

How does the terminology of “social” affect the functioning of the analysed initiatives? (second paragraph of hindering factors). Why one term is perceived as more appropriate?

This term does not affect functioning of the analysed initiatives. However, different understandings (and misunderstanding of “social innovation”) of the term “social” is reflected in e.g. a draft legislation which does not cover the full spectrum of possible social innovations, but narrows “social” down to specific groups of society: disabled people, mentally ill, elderly , Roma people, etc.

Another and negative connotation of “social” is related to the historical background, to the times of the Socialist Federal Republic of Yugoslavia, with centralised government regime under control of the Communist Party. It refers to the then socially-owned (public) enterprises, which were not operating in the interest of private people. This connotation raises reluctance to join social enterprise activities. This issue is reflected in lines 502-514.

Table 1 is very similar to Table 3. Given that the added value of this paper falls in Table 3, and the contents of Table 1 are explained in the text, I recommend the deletion of the first table and rather employ the space for further clarifications.

We thank you for this comment, and we accept it. We deleted table 1 and kept just Table 3 (now table 2 in the text). This table is adapted to the changes we introduced in the discussion section and now it integrates institutional voids and related supporting factors.

The conclusions miss a reference on the informal institutional findings.'

The revised conclusion addresses informal institutions too (lines 832-835).

Reviewer 2 Report

See document attached. 

Author Response

Authors want to thank the reviewers for the time and effort spent on the paper. We have responded to all the comments. Detailed responses to both reviewers are provided in the table below. We hope that with this revision we meet all the requirements and expectations of both editor and reviewers.

We would like to note that full manuscript underwent a professional native speaker proof-reading after we did revision, as requested by one of the reviewers. Attached you will find confirmation of proof-read service.

Reviewer 3 Report

The manuscript is well done. However, I suggest to the author to include a specific section regarding the contribution of this research compared to previous studies.

Besides, the authors should include the manuscript structure in the end of the introduction section.

Author Response

(The authors gave the same response as above.)
